# Analysis of Graviresponse and Biological Effects of Vertical and Horizontal Clinorotation in *Arabidopsis thaliana* Root Tip

**DOI:** 10.3390/plants10040734

**Published:** 2021-04-09

**Authors:** Alicia Villacampa, Ludovico Sora, Raúl Herranz, Francisco-Javier Medina, Malgorzata Ciska

**Affiliations:** 1Centro de Investigaciones Biológicas Margarita Salas-CSIC, Ramiro de Maeztu 9, 28040 Madrid, Spain; avillacampa@cib.csic.es (A.V.); ludovico.sora@mail.polimi.it (L.S.); rherranz@cib.csic.es (R.H.); fjmedina@cib.csic.es (F.-J.M.); 2Department of Aerospace Science and Technology, Politecnico di Milano, Via La Masa 34, 20156 Milano, Italy

**Keywords:** microgravity simulation, gravitropism, gravity perception, plant, clinostat

## Abstract

Clinorotation was the first method designed to simulate microgravity on ground and it remains the most common and accessible simulation procedure. However, different experimental settings, namely angular velocity, sample orientation, and distance to the rotation center produce different responses in seedlings. Here, we compare *A. thaliana* root responses to the two most commonly used velocities, as examples of slow and fast clinorotation, and to vertical and horizontal clinorotation. We investigate their impact on the three stages of gravitropism: statolith sedimentation, asymmetrical auxin distribution, and differential elongation. We also investigate the statocyte ultrastructure by electron microscopy. Horizontal slow clinorotation induces changes in the statocyte ultrastructure related to a stress response and internalization of the PIN-FORMED 2 (PIN2) auxin transporter in the lower endodermis, probably due to enhanced mechano-stimulation. Additionally, fast clinorotation, as predicted, is only suitable within a very limited radius from the clinorotation center and triggers directional root growth according to the direction of the centrifugal force. Our study provides a full morphological picture of the stages of graviresponse in the root tip, and it is a valuable contribution to the field of microgravity simulation by clarifying the limitations of 2D-clinostats and proposing a proper use.

## 1. Introduction

The force of gravity is a permanent environmental factor that exerts fundamental influence on plant growth and development. In particular, the direction of the gravity vector determines the orientation of the plant growth in a process called gravitropism. The gravitropic response is commonly divided into three steps: gravity perception, transduction of the signal, and the growth response [1,2], although some authors split the perception phase into two steps, namely susception—the movement of the statoliths and their sedimentation, and perception—the transfer of the physical act into a physiological signal [3]. Gravity perception takes place in specialized cells called statocytes. These cells have starch-containing plastids, statoliths, which sediment in the direction of the gravity vector [4]. In the root, statocytes are located in the root cap, they are organized in tiers, and form a tissue called columella, whereas in the shoot, they are found in the endodermis [5]. Gravity perception in the root is preceded by the motion of statoliths in the root columella after plant reorientation to sediment according to the direction of the gravity vector. Columella of *A. thaliana* is a highly organized and dynamic structure. It consists of three to four layers of statocytes, sometimes called tiers or stories S1, S2, S3. Statocytes differentiate from the root cap meristem, formed by the columella root cap stem cells (CSC) which are located directly underneath the quiescent center (QC) and above the first layer of statocytes (S1) [6]. CSC activity ensures a constant regeneration of columella cells. The layers that follow (S1–S4) assume first a role in gravity perception, and then they become secretory cells that are important for root tip protection [7,8,9]. The central cells of columella (central cells of the two layers that follow the meristem) are the most important for gravitropic response [10,11,12]. However, statoliths in S3 and S4 tiers, in which statocytes assume secretory functions, do not always sediment after turning the root [11]. It is not completely understood how gravity signal is transduced after statoliths sedimentation on the endoplasmic reticulum (ER) on the cell periphery. One of the most accepted hypothesis is that the physical contact of the statoliths and the cortical ER opens Ca^2+^ channels which creates a Ca^2+^ signal in the cytoplasm [12] that could lead to changes in auxin PIN-FORMED (PIN) transporters polarity [13]. The latest studies confirm that LAZY proteins also play a function in the early stages of gravitropic signal transduction [14,15,16]. It is well reported that auxin is involved in the final stages. According to the Cholodny–Went theory, the differential gradient of this phytohormone results in differential growth. The auxin gradient is formed by differential localization of auxin PIN transporters in response to gravity [17]. The involvement of PIN3, PIN7, and PIN2 in root gravitropic signaling was confirmed by numerous studies [18,19,20]. PIN3 and PIN7, located in the columella cells, change their distribution within minutes after the gravitropic stimuli, next PIN2 transports auxin through epidermis towards the root elongation zone [21,22]. The differential lateral distribution of auxin transport factors in response to a gravitropic stimulus, leads to auxin accumulation at the lower side of the root inhibiting cell expansion in the elongation zone and low auxin levels in the upper side of the root stimulate cell elongation [23]. In effect, non-uniform growth in the elongation zone enables the root to bend towards the new direction of the gravity vector.

Whereas many studies have been devoted to investigating the plant response to gravitropic stimuli, less is known about the mechanisms triggered by the plant in the absence of any gravity vector or when the magnitude of this vector is significantly lower than the Earth gravity, 1 g. This knowledge is necessary for any enterprise of space exploration, since the environment of outer space and of the nearby planets and satellites is characterized by a low or near-zero gravity force (microgravity). The experiments performed on the orbit in the International Space Station (ISS) provide the best conditions for real microgravity research, but are limited by the high cost and complex logistics. This has motivated scientists to look for more accessible ways to investigate the response of an organism to the low gravity levels. There are a number of hardware devices (also called Ground Based Facilities, GBF) that enable weightlessness simulation with the objective to prevent an organism from perceiving the gravity vector (reviewed in [24]). One of the first created simulators are clinostats that were originally designed to be used with plants [25] but later were also applied with different systems, as for example, in vitro cultures. In clinostats, weightlessness simulation is achieved by averaging the gravity vector during each cycle of rotation. It was designed to be used in plants due to the relatively long presentation time for the gravistimulus (the time that the stimulus must persist to trigger a gravitropic response) in comparison to animal models [26,27,28]. In *A. thaliana*, the presentation time is approximately 0.4–1 min [7,11,29]. In the perfect clinorotation settings the sedimentation fall of the statoliths is converted into quasi-circular paths by continuously rotating the whole plant and in effect the position of the statoliths remain virtually stationary within the cells [30]. It should be noted that, to perform an optimal simulation on a clinostat, the mechanism of graviperception of a given organism has to be taken into account. Despite the common use of clinostats and the vast research that was performed on these devices, no clear set of rules has been established as for the optimal clinostat settings. Angular velocity varies between studies from 1–2 rpm up to 60 rpm without a justification [31,32,33,34,35]. In addition, we can distinguish two types of clinorotation depending on the orientation of the sample in relation to the clinorotation axis: vertical clinorotation (VC), with the longitudinal growth axis of the plant perpendicular to the rotation axis, or horizontal clinorotation (HC), with the longitudinal seedling axis parallel to the rotation axis (see Figure 1) [36]. The possible impact of either of these two types of clinorotations was compared by [36,37,38], concluding that vertical and horizontal clinorotation result in different outcomes. Additionally, different effects of the centrifugal force on plant growth in horizontal and vertical orientation were reported by [39], but there is not a consensus or evidence-based guideline in the best practice to use clinorotation in terms of speed, orientation of the Petri dishes containing samples and the maximum duration of the treatment.

Here, we explore *A. thaliana* responses to fast and slow horizontal and vertical clinorotation, by investigating the elements of each of the three gravitropism stages. We conclude that the seedling growth is modulated differently in fast and slow clinorotation as well as in vertical and horizontal. We have observed directional growth in fast clinorotation and differences in the statolith distribution in different clinorotation conditions. Finally, we observe increased stress response in columella cells and the meristem in horizontally clinorotated seedlings, that does not seem to be related to the gravitropic perception, but rather to enhanced mechano-stimulation. Our results confirm preliminary findings and a mathematical model described previously [40].

## 2. Results

### 2.1. Angular Velocity and Sample Orientation Influence Root Growth Direction and Rate

First, we analyzed root growth direction and rate after 24 h of fast or slow, horizontal or vertical clinorotation. In Figure 1, the sample orientation and acting forces in vertical or horizontal clinorotation are explained.

Seedlings in the control group continued to grow vertically downwards with minor fluctuations. The directional growth control group after reorientation grew downwards according to the direction of the gravity vector (Figure 2a). In the slowly clinorotated samples roots turned slightly, but their direction did not reflect the direction of the centrifugal force vector suggesting it has a negligible magnitude at 1 rpm in our experimental settings. On the other hand, as shown in Figure 2a (and Appendix A), at 60 rpm, the centrifugal drift had a significant impact on the direction of the root growth. As a result, a clear directional growth towards the outside of the Petri dish was observed in the seedlings that grew on the laterals at high angular velocity in both orientations (Figure 2a) due to high acceleration which increases with the distance from the clinorotation axis. The direction and the angle of the roots corresponded to the direction of the centrifugal force vector (black arrows in Figure 2a) and the angle of the root depended on the distance of the seedling from the clinorotation axis (Figure 2a). The plot correlation between the distance of the seedling to the center of rotation and the root tip angle is shown in Appendix A. It should be noted that the roots did not grow straight downwards in any of the conditions, meaning the root growth toward the gravity vector was inhibited by clinorotation. Nevertheless, in case of the fast clinorotation, centrifugal drift seemed to be perceived by the seedlings placed at a distance from the clinorotation axis (Appendix A).

The maximum centrifugal force for both configurations, vertical and horizontal, at slow clinorotation (1 rpm) across the entire Petri Dish (4.5 cm radius) was negligible (up to 5.04 × 10^−5^ g at the border of the Petri dish). In turn, the theoretical value of centrifugal acceleration in the fast clinorotation (60 rpm) is 0.18 g at 4.5 cm radius, which is above the graviperception threshold values reported before [41,42,43]. These calculations are in agreement with the growth pattern observed in the experiment. 

We have observed that in the Horizontal Fast Clinorotation (HFC), the seedlings were often displaced after 24 h (see Figure 2a; the position of the seeds before the clinorotation is marked with short black lines along the line in the middle of the Petri dish). Sections of the seedlings detached from the substrate. It should be noted, that in the experimental setting when seedlings are grown on an agar surface on a Petri dish, in vertical clinorotation, the gravity vector acts along the surface and so does the centrifugal force (both acting in the same plane). On the other hand, in horizontal clinorotation, although the centrifugal force acts along the agar surface, the Petri dish changes constantly the angle with respect to the gravity vector, meaning that the gravity vector and the centrifugal force do not act in the same plane, except at the time when the Petri dish is positioned vertically (see Figure 1). This constant change causes the seedlings to partly detach from the agar. It is possible that in horizontal clinorotation the “change of phase” has an additional impact on the seedlings´ response. In this experimental setting, the gravity vector acts during half of the clinorotation cycle by “pushing” the seedling against the agar surface (agar side down) and during the other half of the cycle by “pulling” the seedling from the agar surface (top of the Petri dish down). As mentioned before, in vertical clinorotation this issue is not present, since both the gravity vector and the centrifugal force act along the agar surface.

To measure the influence of the speed and the orientation of the sample on the root growth rate we measured the length of the roots before the clinorotation (point 0 marked with arrows in the Figure 2a) and after 24 h (L, see Figure 2b). We have observed a significantly higher root growth rate in the HFC sample (Figure 2c).

We analyzed in more detail the root morphology using four values described before by [44], namely, integral averaged angular declination (α in Figure 2b) of the root tip, vertical growth index (VGI), horizontal growth index (HGI), and root straightness, later renamed as gravitropic index (GI) [20,45,46,47,48]. VGI is defined as the ratio between the straight line distance from the base of the root to the root tip and the root length. The closer VGI value is to 1, the straighter the root grows. On the other hand, HGI is the distance in the horizontal line between the base of the root and the root tip divided by the root length. The higher the deviation from the straight line, the higher is the value of the HGI (closer to 1). Here, we calculated horizontal and vertical growth indexes taking into account the distance from point 0 (start of the clinorotation) to the root tip after 24 h (Figure 2b).

The GI is the shortest distance from the base of the root to the root tip divided by the root length. Similar to the VGI, the closer the value is to one, the straighter is the root.

As expected, in the control the HGI was the lowest and VGI the highest and in the directional growth control the trend was opposite (Figure 2e,f). As indicated by HGI, VGI, and α values, roots of the seedlings exposed to vertical clinorotation deviated less from a straight line than in case of horizontal clinorotation (Figure 2d–g). Although the differences between the values that correspond to the different clinorotation conditions were not significant, the trend was repeated in all parameters, including the plot of correlation between distance from the rotation axis and the angle of the root (Appendix A).

### 2.2. Distribution of Statoliths in the Columella Is Influenced by Angular Velocity

We have investigated the distribution of statoliths in the columella in Differential Interference Contrast (DIC) images of formaldehyde (FA) fixed seedlings (Figure 3 and Appendix A).

Taking into account the importance of each tier in the columella, we have focused on central statocytes (in S1 and S2 tiers) to compare the position of statoliths inside the statocytes in each condition. These are outlined in Figure 3.

Statoliths containing starch grains could be seen within statocytes in the DIC images. Although other organelles and internal cellular structures, such as the nucleus or ER, could not be distinguished, the contour of the statocytes could be observed. In the central statocytes of columella in the control roots, groups of statoliths could be seen located close to the bottom of the cell (Figure 3, blue arrows in the control sample). In the directional growth control, statoliths were clustered, close to one lateral cell wall of the statocyte, at 1, 2, and 3 hours after the turn (Figure 3, red arrows). After 24 h, the statoliths were located in the lower part of the statocytes when the root has already changed its orientation and grew downwards, according to the gravity vector. In the slow clinorotated samples, Vertical Slow Clinorotation (VSC) and Horizontal Slow Clinorotation (HSC), groups of statoliths were observed more dispersed throughout the statocytes in comparison to the controls and samples exposed to Fast Clinorotation (FC) for 3 and 24 h (blue arrows in Figure 3 in VSC and HSC samples). In the fast clinorotated samples, statoliths formed groups located in the lower part of the cell (blue arrows in Figure 3 in Vertical Fast Clinorotation (VFC) and HFC samples). This pattern was more evident in the VFC than in HFC, in which some statoliths could be seen in the center or even in the upper part of the cell (Figure 3). Although the fixation of the samples was performed immediately after clinostat was stopped (stationary mode), the differences observed between the conditions confirm that the minimal time of this procedure did not significantly influence the position of the statoliths.

### 2.3. Statocytes’ Ultrastructure Is Affected by Horizontal Clinorotation and Displays Features of Stress Response

We investigated in more detail the cell ultrastructure of central statocytes in S1 and S2 tiers, in different conditions, by electron microscopy. Statocytes show a polar distribution, meaning that the nucleus is located in the upper part and the statoliths in the lower part of the cell. We have observed this typical layout in all the conditions (Figure 4a). The round nucleus, enclosed in a double nuclear membrane, did not present any abnormalities in any condition. Near the nucleus, numerous mitochondria and lysosomes were located. Mitochondria presented elongated (oblong) shape with compact cristae in cross sections, although round mitochondria with more loosely organized cristae were also observed in HSC (Figure 4b). In controls and in vertically clinorotated samples, the outline of lysosomes showed round shape, whereas in HSC sample their shape was more oblong (Appendix A). In fast clinorotated samples the size, and in HFC the number of lysosomes was reduced. A complex ER system could be observed in all the conditions in the lower part of the cell and also in a simpler form on the laterals and close to the nucleus. Multiple Golgi bodies were present distributed around the cell, which confirmed high secretory activity. Statoliths containing multiple starch grains and sometimes fibrous bundles [49] could be observed in the central and lower part of the cell and in the sample turned 90°, also in the upper part in proximity to the nucleus. Statoliths were of regular round or oblong shapes, although in the directional growth control (90°) and HFC, irregular shapes were observed (Figure 4a). Occasional vacuoles were observed in the controls and vertically clinorotated samples. In horizontally clinorotated samples, the vacuoles were bigger and numerous in the HSC sample. Cell walls in all conditions could be easily distinguished between plasma membranes of adjoining cells and middle lamella was sometimes visible (Figure 4c). The walls were mostly thin, although thicker regions could be also seen (Figure 4a). Plasmodesmata connecting cells from different tiers (up/down) could be often seen, suggesting close communication between these cells. Vesicles (endosomes) were often observed between the cell membrane and the cell wall. In controls and vertically clinorotated samples, the cell walls delimited mostly straight rectangular cells; nevertheless, in horizontally clinorotated samples, the anticlinal and bottom cell walls had often irregular wavy shape, especially pronounced in the HSC sample (Figure 4a,c; Appendix A) which could suggest they underwent prolonged mechanical stress.

### 2.4. The Distribution of PIN2, an Auxin Transporter, Is Affected in the Horizontal but Not Vertical Clinorotation

Auxin is one of the most important phytohormones, and it plays an essential role in regulating root growth. In the meristem, auxin promotes proliferation (mitosis), whereas in the elongation zone, it inhibits cell expansion [50,51]. It is transported from the shoot to the root meristem through the central part of the root and an auxin maximum is formed around the quiescent center. From there, auxin is transported, first by PIN3 and PIN7, and then by PIN2 proteins, up to the elongation zone through epidermal cells by basipetal transport, sometimes called the reflux loop [17,22,52]. This transport is responsible for asymmetrical changes in auxin gradient that modulate cell expansion in the elongation zone, enabling the root to bend in response to reorientation. We have investigated changes in auxin transport in the PIN2-Green Fluorescence Protein (PIN2-GFP) reporter line (Figure 5), as well as auxin levels and distribution, using DR5-β-glucuronidase (DR5-GUS) and DII-Venus reporter lines exposed to different clinorotation conditions (Figure 6 and Appendix A, respectively).

In the directional growth control (90°) we observed asymmetric changes in PIN2-GFP distribution, characteristic for gravitropic response, with accumulation on the lower side. This distribution was especially conspicuous 2 h after the reorientation (Figure 5a, red arrow), whereas in the control sample the distribution was symmetric at all times. In the clinorotated samples the asymmetric distribution was not observed, although in the horizontal clinorotation we have observed PIN2 internalization in the lower endodermis, that was confirmed by fluorescent signal quantification in the different meristematic layers: epidermis, cortex, and endodermis (Figure 5a,c). To investigate if this distribution is related to auxin transport inhibition, we have grown PIN2-GFP seedlings on medium complemented with naphthylphthalamic acid (NPA), an auxin transport inhibitor that inhibits gravitropic response [23], at different concentrations (Figure 5b). Indeed, the PIN2 accumulation in the lower endodermis was observed in NPA treated seedlings, suggesting that the observed distribution is a result of auxin transport inhibition (Figure 5b,d). The samples horizontally clinorotated for 24 h showed PIN2 intracellular accumulation, similar to that in the sample treated with 1 µM NPA (Figure 5). The typical polar distribution of PIN2 in the meristematic epidermis and cortex was not affected in NPA-treated plants, in agreement with previous reports [53].

Next, we investigated auxin distribution and levels during clinorotation in DR5 and DII-Venus reporter lines (Figure 6 and Appendix A). DR5 is a synthetic auxin response element that is highly responsive to auxin level changes. We have used a fusion of DR5 and a minimal 35S Cauliflower mosaic virus (CaMV) promoter with GUS reporter gene in DR5-GUS reporter line [54]. DR5 (TGTCTC) binds to auxin response factors and responds rapidly to active auxin concentration between 10^−8^ and 10^−5^ M and remains at high levels up to 10^−4^ M [55]. This makes it a suitable tool for tracking the accumulation of auxin in combination with the GUS reporter gene.

DII-Venus is a more modern auxin sensor, which combines VENUS fast maturing form of yellow fluorescent protein and Aux/**** Indole-3-acetic acid (IAA) auxin-interaction domain (domain II; DII) expressed under a constitutive promoter [56]. It enables tracking of Aux/IAA-dependent degradation of VENUS fluorescent protein. This provides a tool to monitor dynamic changes of auxin levels that does not depend on a complex auxin response pathway, as is the case of DR5. DII-Venus is constitutively expressed in nuclei and it is degraded upon contact with auxin, resulting in loss of fluorescent signal and directly reflecting auxin levels in the cell [56].

The typical auxin distribution in the root meristem with an auxin maximum around the region of quiescent center and diminishing levels towards the elongation zone, was observed in all conditions. In the directional growth control (90°) we have observed asymmetric auxin distribution at the meristem laterals after 1–3 h after the reorientation (Figure 6, red arrows). No obvious changes in the auxin distribution pattern was observed in any of the clinorotation conditions.

We have also grown DII-Venus and DR5-GUS seedlings on a medium supplemented with NPA at different concentrations, to investigate whether the distribution of PIN2-GFP corresponding to auxin transport inhibition is reflected in auxin accumulation in these reporter lines in any of the different clinorotation conditions. The seedlings treated with NPA showed gradual accumulation of GUS signal in the central and lateral parts of the root tip in DR5-GUS line and gradual decrease in DII-Venus signal that depended on NPA concentration (Figure 6b, d, and Appendix A). Nevertheless, a similar pattern was not observed in any of the clinorotated samples (Figure 6a and Appendix A).

We quantified DR5 relative staining to investigate the changes in auxin levels in different conditions. The only statistically significant difference was a minor reduction in auxin levels after 2 h of HSC in comparison to control (Figure 6c). On the other hand, a clear increase in GUS signal was observed in NPA-treated seedlings.

## 3. Discussion

The clinostat is an important tool for investigating the graviresponse of plants and, in particular, the impact of microgravity, thus being a useful complement to space experiments. It was designed to be applied in plants for their slow gravitropic response [26,27,28] but nowadays is also used in experiments with animals and in vitro cell cultures [57,58,59,60,61,62]. Since the first use of clinostat in plant studies [25,27,28] gravitropism was extensively studied and today is much better understood. The response of the plant to reorientation is a dynamic sequence of processes that leads to a non-uniform growth in the elongation zone in approximately 3 h, according to the direction of the gravity vector [23,63]. The three phases of gravitropism can be distinguished: gravity perception in the columella, signal transduction that results in auxin distribution gradient, and non-uniform elongation that leads to the root bending. Each phase takes a certain time, but the gravity perception, which is the key step to take into account for an optimal microgravity simulation, takes approximately 6 minutes (370 s after reorientation) [12].

Although clinostats are widely used, little bibliography is available on comparison of fast and slow [64], or vertical and horizontal clinorotation [36,37,38,65]. In this respect, it is worth mentioning that we have previously developed a mathematical model which enabled calculation of the clinostat setting for an optimal microgravity simulation. Given the size of the experimental container, optimal angular velocity can be calculated as a function of the time of clinorotation, and vice versa [40].

The present study confirms that all clinorotation conditions are enough to avoid the root growth according to gravity vector. Nevertheless, in fast clinorotation, centrifugal force led to conspicuous directional growth, as will be discussed further. In the slow clinorotated samples, non-directional, random root growth was observed, similar to the growth observed in plants grown in real microgravity conditions [66,67]. A study in the International Space Station showed that the orientation of roots in microgravity was not random, but was the result of automorphogenesis and autotropism, successively. First, the embryonic root curved strongly away from cotyledons, and then it grew straight [42]. As shown in the DIC images of root columella, under slow clinorotation, statoliths appeared dispersed in statocytes similar to what was observed in plants grown in real microgravity [36,68,69,70,71,72].

Clear directional growth was triggered at high angular velocities by centrifugal force in the seedlings placed furthest away from the clinorotation axis. The value of this force is proportional to the distance from the clinorotation axis, meaning that the plants positioned further from the center of the Petri dish perceived the centrifugal force and responded to it. Thus, fast clinorotation would only be appropriate in a very limited radius from the center of rotation, or for short times of microgravity simulation. Previously, researchers applied the criterion of centrifugal force limit to determine the usable space for effective microgravity simulation during clinorotation [31,38,73,74]. These limits ranged between 0.00009 g – 0.2 g and were determined by investigating the minimal centrifugal force that caused directional growth in oats clinorotated horizontally (0.0001 g) [41], in lentils in a centrifuge in the GRAVI-1 space experiment (0.000014 g) [42], or the partial- g effect on rhizoids of *Chara globularis* in parabolic flights (0.05 g) [43]. In our experimental setting, at 1 rpm, the centrifugal force is below most of the reported perception thresholds (at the edge of the Petri dish; 4.5 cm from the clinorotation center, 0.00005 g) which was confirmed by our morphological study. On the other hand, at 60 rpm the centrifugal force (at the edge of the Petri Dish; 4.5 cm from the clinorotation center, 0.18 g) exceeds the graviperception thresholds at a short distance from the clinorotation axis and triggers directional growth, as shown in the morphological study.

Taking into account that the directional growth was not observed in slow clinorotation our morphological study suggests that slow clinorotation is more suitable for microgravity simulation in *A. thaliana* seedlings. This is in agreement with previous reports indicating that angular velocity values between 0.33 and 2 rpm result in effective clinorotation (microgravity simulation) [30,38,75], and with our preliminary results [40].

We can distinguish vertical or horizontal clinorotation depending on the orientation of the sample in relation to rotation axis [36]. Previous studies have shown that the type of the clinorotation (vertical or horizontal) influences the plant response in a different way. John and Hasenstein [38], have demonstrated that horizontal clinorotation (1–5 rpm) is less effective in nullifying gravitropic signals than vertical clinorotation. Seedlings were turned 90° before clinorotation for up to 15 min and then clinorotated vertically or horizontally. Seedlings that were horizontally clinorotated showed more pronounced directional growth than the vertically clinorotated ones. Lorenzi and Perbal [36], compared cell ultrastructure and concluded that VSC has similar effect on the position of the nucleus as real microgravity.

Moreover, a number of studies pointed out that horizontal clinorotation is associated with higher stress response [49,65,69,73], although this aspect was later extrapolated as a general effect of clinorotation. This observation is in agreement with our results, as we observed altered ultrastructure of statocytes (Figure 4) and PIN2 intracellular accumulation (Figure 5) in horizontally clinorotated samples. The ultrastructural changes we report, especially pronounced in HSC, such as increased vacuolar compartment, changes in outline shape and ultrastructure of mitochondria, and irregular cell wall shape, were postulated to be linked to the response to abiotic stress in *A. thaliana* and *Pisum sativum* [76,77]. Additionally, seedlings of *P. sativum* exposed for 3 days to horizontal clinorotation showed increased levels of stress indicators, such as heat shock proteins HSP70 and HSP90 [65]. Increased lipid breakdown was observed in rapeseed (*Brassica napus*) seedlings after HSC (1 rpm) for 5 days. Columella degradation, as well as ultrastructural alterations, in agreement with our result of irregular cell wall shape and increased lytic compartment, were reported for horizontally clinorotated *Lepidium sativum* seedlings for 20 h [73]. In addition, in two-axis clinostat experiments, where the sample is clinorotated in both horizontal and vertical orientations, columella degradation, as well as increased lytic compartment and irregular cell wall shape, were observed in white clover (*Trifolium repens*) clinorotated for 72 h [49,69]. These alterations were not observed in real microgravity [49,69], which suggests that they could be an artifact of the simulation and not due a microgravity effect. This stress response was not observed in vertical clinorotation in our study.

PIN2 is expressed in root epidermis and cortex under normal conditions, a pattern that was observed in control and vertically clinorotated seedlings. Samples exposed to horizontal clinorotation for prolonged periods of time (3 h and more) additionally presented intracellular localization in the endodermis. PIN2 localization in the endodermis was previously described in seedlings with disrupted symplastic signaling to and from the endodermis [78] and in mutants deficient in Calcium-Dependent Protein Kinase-Related Kinase 5 (CRK5) protein, a plasma membrane-associated kinase which phosphorylates the hydrophilic loop of PIN2 [79]. Nevertheless, the endodermal localization of PIN2 in both cases was associated with diminished levels of PIN2 in epidermis and endodermis, which was observed in NPA-treated seedlings, but not in horizontally clinorotated samples. Intracellular localization of PIN2-GFP in lytic vacuoles in the epidermis was previously described in seedlings treated with concanamycin A (an inhibitor of vacuolar proton ATPases which reduces protein degradation) and after incubation in the dark for 6 h [80]. Nevertheless, we only observed the intracellular PIN2 localization in the endodermis. Endodermis-deficient *sgr* (*shoot gravitropism*) mutants were deficient in shoot gravitropism but presented normal root gravitropism, suggesting that this layer is not crucial for root gravitropism [5,81]. The mechanism and function of the intracellular PIN2 localization in the endodermis under prolonged horizontal clinorotation are not clear but might be related to the intensification of mechanical stimuli.

Although reports of increased stress response to horizontal clinorotation are substantial, it is not well understood what is the nature of the stress. A suggestive and feasible possibility is that this stress would be related to the thigmotropic reaction. In vertical clinorotation, the gravity vector and centrifugal force act on the same plane, along the agar surface; therefore, thigmotropic stimuli are constant. On the other hand, in horizontal clinorotation, the position of the Petri dish with respect to the gravity vector changes constantly, meaning that the Petri dish is tilted at different angles (from 1° to 360°) during the clinorotation cycle. In fact, it is well established that seedlings grown on a tilted agar surface present altered growth patterns; waving, coiling, and skewing [82,83,84]. A plausible explanation for this phenomenon is that, since the root is not able to penetrate agar surface, it is perceived as an obstacle and activates the obstacle avoidance mechanism [85]. This mechanism is regulated by the columella [85,86]. While thigmotropic reaction is activated, gravitropism is attenuated [85]. Although the mechanism of obstacle avoidance is not well understood, it was recently reported that auxin and Ca^2+^ are involved in this process [86,87], factors that are also key players in gravitropism. Obstacle avoidance is fulfilled by two acts of root bending, with the first bend appearing just 20 minutes after the contact with an obstacle [87]. This suggests that thigmotropic growth response is faster than the gravitropic response, in which asymmetrical growth appears approximately 3 h after reorientation [23,63]. It is possible that, during the HSC, the repetitive cycles when the Petri dish is tilted (agar down—more friction, top of the Petri dish down—no friction) could be enough for the root to detect agar surface as an obstacle and trigger the thigmotropic response. In effect, this response would be triggered in each clinorotation cycle and could lead to the accumulation of mechanical stimuli and stress response. This hypothesis is supported by the fact that in HFC, where the clinorotation cycles are shorter, severe ultrastructural changes in statocytes were not observed, which could suggest that the duration of the mechanical stimuli was under the limit of the thigmotropic perception.

Revisiting the case of the Random Position Machine (RPM), this device is particularly interesting as a combination of HFC and VFC, usually involving fast angular speed and sudden changes in direction caused by the operation in real random mode [88]. Actually, centrifugal forces and the experimental design requirement of aligning seedling growth with the rotating axis become irrelevant in this simulator, due to forces averaging in the three dimensions. However, it would be conceivable that the overall stress response in the statocytes could be even higher than the one reported here. Our comparisons of RPM experiments with real microgravity and low g levels (< 0.1 g) produced by centrifugation in orbit [89,90] suggest that the seedlings may respond in the RPM, not only to microgravity alterations, but also to a certain misbalance in the different tropism signals (gravity, light, touch, water, etc.) that is also observed in partial gravity conditions recreated in orbit. Otherwise, clinostat and RPM produced comparable results with *A. thaliana* seedlings in a parallel study on the effects of simulated microgravity on root meristematic cells [32].

To sum up, different clinostat settings can produce different forces in combination with the gravity force. One of the drawbacks of clinostats and other ground based facilities used for microgravity simulation is the fact that most studies only deal with the gravitropic response, when other tropisms, such as thigmotropism, may apply and should be taken into account. Considering that the mechanisms regulating these responses are still not well-understood any simulator may introduce artefacts. Further investigation of the plant physiological response to different simulation conditions and its comparison to the response to real microgravity will help us to discern the effect of microgravity from other aspects of the mechanical simulators and help us to improve the simulation quality.

## 4. Materials and Methods

### 4.1. Material, Growth Conditions, and Quantification

The 2D-clinostat was granted to our laboratory by the Zero-Gravity Instrument Project (ZGIP, United Nations Office for Outer Space Affairs (UNOOSA)). An adaptor, designed and constructed ad hoc with a 3D-printer, was used for horizontal clinorotation (Appendix A). The two orientations in the use of the clinostat are shown in Figure 1. Horizontal clinorotation with the seedlings parallel to the rotation axis (as shown Figure 1b) was previously applied in multiple studies [31,32,34,35,91].

Seeds were surface sterilized with 70% ethanol with 0.01% Triton X-100 followed by 95% ethanol for 3 minutes and air-dried. In total, 7 seeds were positioned in the middle of the 9 cm Petri dish. Seedlings were grown on Murashige and Skoog (MS) medium (M0221, Duchefa) with 0.5 g/L 2-(*N*-morpholino)ethanesulfonic acid (MES) (M8250 Sigma–Aldrich), 1% sucrose (107651, Merck), and 0.8% plant agar (P1001, Duchefa) for 5 days at 23 °C under long day regime (16 h/8 h) vertically for 5 days to obtain straight root growth according to the gravity vector. After 5 days, seedlings were clinorotated for 24 h on a clinostat horizontally (H) or vertically (V) at two speeds; at 1 rpm (slow clinorotation, SC) and at 60 rpm (FC). Parallel to the experiment we performed two controls, one kept in the vertical position (1 g control) and another one turned 90 degrees and kept in vertical position (directional growth control). Samples were grown in darkness under the four conditions. A total of three runs were performed for horizontal and vertical clinorotation (position of the plants in respect to the rotation axis is presented in Figure 1) for each experiment. In total around 21 seedlings of each sample were analyzed in each experimental procedure. Pictures were taken before and after 1, 2, 3, 24 h of exposure to each condition. Root length and growth during 24 h of clinorotation and in controls (Figure 2a) were measured with ImageJ software. Vertical and horizontal growth indexes and the integral averaged angular declination (α) were calculated from time 0 as described in [44] (Figure 2b). GI was determined from the base of the root to the tip as in [44].

For NPA treatment, seedlings were grown on medium complemented with 0 (0.01% (v/v) Dimethyl sulfoxide (DMSO)), 1 or 5 µM NPA in DMSO for 5 days.

### 4.2. Optical and Electron Microscopy

For microscopical analysis, samples were fixed immediately after the clinorotation, taking care of minimizing the time elapsed between the arrest of the clinostat and the interaction of the fixative with samples. First, the fixative was directly added to the Petri dish just after the release from the clinostat, for an immediate arrest of the vital activity. Then, seedlings were transferred to centrifuge tubes filled with fixative solution. For GUS staining, DR5-GUS seedlings [54] (seeds kindly supplied by Dr. E. Carnero-Diaz, Sorbonne University, Paris, France) were fixed in 90% acetone at −20 °C for 12 days. Seedlings were washed 3 times with 0.1 M sodium phosphate buffer pH 7.2 and GUS signal was revealed by enzymatic reaction with 1 mM X-GlcA (X1405, Duchefa), 1 mM potassium ferricyanide (P4066, Sigma–Aldrich), 0.25 mM trihydrate ferricyanide (455989, Sigma–Aldrich) in 0.05 M sodium phosphate buffer pH 7.2, at 37 °C overnight. Samples were washed 3 times in 0.05 M sodium phosphate buffer pH 7.2, mounted in 85% glycerol and observed under a Leica DM2500 microscope. Staining intensity was quantified with ImageJ in the meristem area in grey scale in the saturation channel (HSB Stack) [40].

For electron microscopy the *A. thaliana* seedlings were fixed in 2.5% glutaraldehyde (GA) and 1.5% formaldehyde (FA) and processed as previously described in [92]. Samples were embedded in LR White resin (London Resin, Berkshire, UK).

### 4.3. Confocal Microscopy

For confocal microscopy, Wild Type (WT) (Col0) and DII-Venus [56] (Nottingham Arabidopsis Stock Centre (NASC) ID: N799175) seedlings were fixed in 3% formaldehyde in Phosphate-buffered saline (PBS), as described in paragraph 4.2., for 1 h at room temperature (RT). Next samples were washed three times in PBS and digested for 1 h in 0.1% pectinase (17389, Sigma–Aldrich, St. Louis, MO, USA), 0.5% macerozyme (16419, Serva, Heidelberg, Germany), 0.4% mannitol (105983, Merck), 10% glycerol, and 0.02% Triton X-100 in PBS at 37 °C. Next, seedlings were washed 3 times with 10 % glycerol and 0.02 % Triton X-100 in PBS. PIN2-GFP seedlings [22,93] were processed as above but substituting PBS for Microtubule Stabilizing Buffer (MTSB) [94]. Seedlings were mounted onto poly-L-lysine coated multi-well microscope slides and dehydrated with a drop of 90% acetone in each well. For cell wall staining, samples were first permeabilized with 1% NP-40 (I8896, Sigma–Aldrich, St. Louis, MO, USA) and 0.5% Deoxycholic acid (DOC) (D2510, Sigma-Aldrich, St. Louis, MO, USA) in PBS and then stained for 2 h with 2% SCRI Renaissance Stain 2200 [95,96] (Renaissance Chemicals, North Duffield, UK) with 4% DMSO (81802, Sigma–Aldrich, St. Louis, MO, USA) protected from light. Finally, samples were washed twice with PBS and mounted in 1,4-Diazabicyclo[2.2.2]octane (DABCO) (D2522, Sigma–Aldrich).

Images were obtained with a confocal microscope Leica TCS SP5 with Acousto Optical Beam Splitter (AOBS) using 63 X or 40 X oil immersion optics. Yellow Fluorescence Protein (YFP) (DII-Venus) and GFP (PIN2-GFP) were excited at 496 nm and SRCI Renaissance Stain 2200 at 405 nm with Argon and UV lasers, respectively. Pseudocolor reflecting the intensity of the GFP and YFP signal was applied with the Lookup Table Royal Tool in the ImageJ software. Relative fluorescence intensity in the different meristem layers was quantified using ImageJ, selecting the area with the SCRI Renaissance Stain 2200 channel and quantifying the intensity in grey scale in the GFP channel. Differential Interference Contrast (DIC) images of columella were taken using Leica TCS SP5 microscope to observe the localization of the statoliths within statocytes.

### 4.4. Statistics

In order to analyze statistical differences, SPSS software was used. Data from 3 independent experiments (7 seedlings for each condition in each experiment, in total 21 seedlings per condition) were analyzed. Normality was tested with Kolmogorov–Smirnov test, and homocedasticity with Levene test. Statistical differences were tested with ANOVA (root growth), ANOVA with Welch correction (HGI and PIN2-GFP fluorescence in meristematic layers) or U-test Mann–Whitney Wilcoxon (VGI, GI, root angle, and DR5-GUS staining) upon normality and homocedasticity test results.

## 5. Conclusions

Our results confirm that the plant responds differently to vertical and horizontal clinorotation and that the latter is related to the stress response, which is especially evidenced in horizontal slow clinorotation. This stress response is not present in vertical clinorotation. In fast clinorotation directional growth is triggered by centrifugal force which is proportional to the distance from the clinorotation axis. Taking into account that the stress response is not observed in the slow vertical clinorotation and that the growth pattern and the statolith distribution in this condition are similar to the ones observed in real microgravity experiments, we conclude that VSC is the least susceptible to artefacts related to microgravity simulation.

## Figures and Tables

**Figure 1 plants-10-00734-f001:**
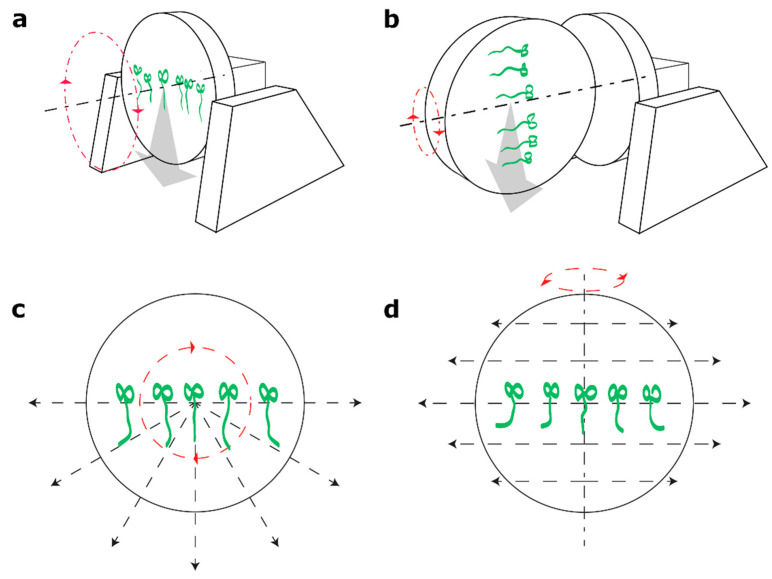
2-D clinostat and sample orientation. Schematic representation of (**a**) Vertical clinorotation, (**b**) Horizontal clinorotation, (**c**) Theoretical forces and direction in vertical clinorotation, and (**d**) Theoretical forces and direction in horizontal clinorotation. Red dashed arrows—the direction of clinorotation; black dashed arrows—the direction of the centrifugal force; gray arrows —direction of the gravitational force.

**Figure 2 plants-10-00734-f002:**
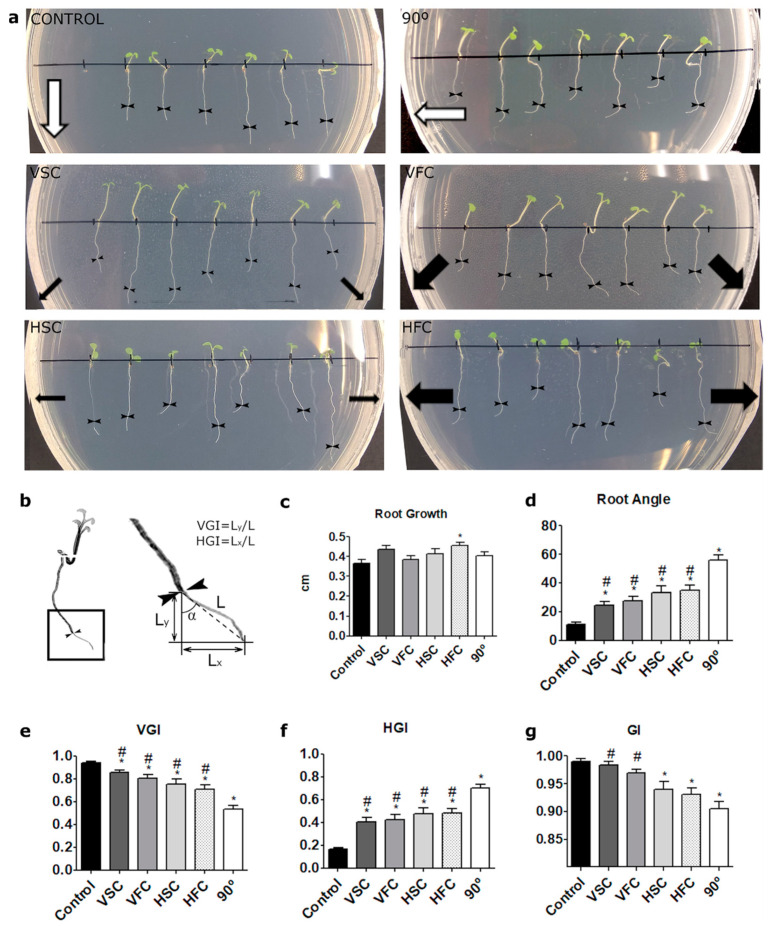
Root growth and direction upon clinorotation conditions. (**a**) Photographs of seedlings in the different experimental conditions (CONTROL, control; 90°, directional growth control; VSC, Vertical Slow Clinorotation; VFC, Vertical Fast Clinorotation; HSC, Horizontal Slow Clinorotation; HFC, Horizontal Fast Clinorotation). White arrows represent gravity direction and black arrows represent the direction and magnitude of centrifugal force (narrow arrow for low magnitude in slow clinorotation and wide arrow for high magnitude in fast clinorotation). (**b**) Schematic representation of root growth parameters quantified (**c**–**g**) Quantification of different root features: (**c**) the root growth, expressed as the length of the root from time point 0 (arrows) and after 24 h; (**d**) root angle, expressed as the absolute value of integral average angular declination; (**e**) VGI, Vertical Growth Index; (**f**) HGI, Horizontal Growth Index; (**g**) GI, Gravitropic index. * *p*-value < 0.05 compared to the control. # *p*-value < 0.05 compared to directional growth control, 90°.

**Figure 3 plants-10-00734-f003:**
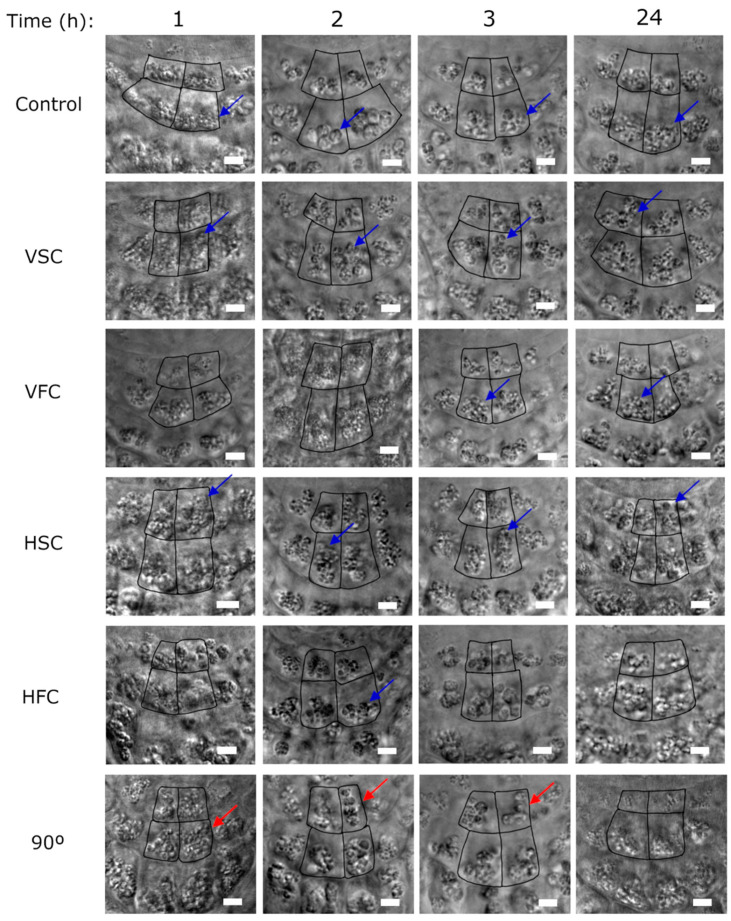
Statoliths position in Differential Interference Contrast (DIC) images. Detail of central S1 and S2 statocytes in the root columella at the different times (1, 2, 3, and 24 h) and conditions (Control; VSC, Vertical Slow Clinorotation; VFC, Vertical Fast Clinorotation; HSC, Horizontal Slow Clinorotation; HFC, Horizontal Fast Clinorotation; 90°, directional growth control). Blue arrows highlight statolith differential position among conditions. Red arrows indicate statoliths position changes due to the gravity vector in the directional growth control. Scale bar represents 5 µm2.

**Figure 4 plants-10-00734-f004:**
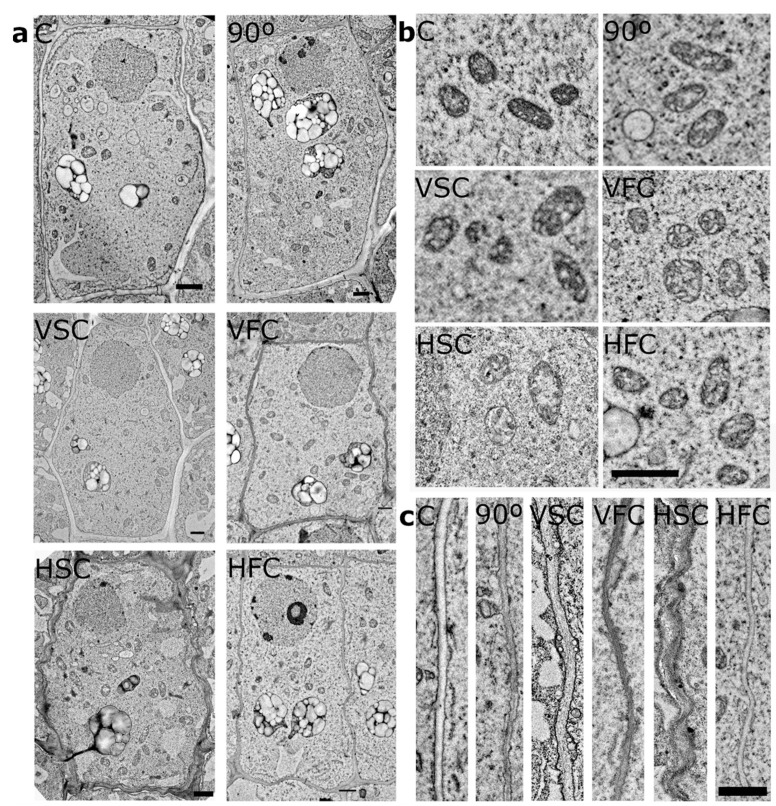
Statocytes ultrastructure. (**a**) Electron microscopy images of representative statocytes of S2 central columella tier in the different experimental conditions: Control (C), directional growth control (90°), Vertical Slow Clinorotation (VSC), Vertical Fast Clinorotation (VFC), Horizontal Slow Clinorotation (HSC), and Horizontal Fast Clinorotation (HFC). (**b**) Detail of mitochondria structure. (**c**) Detail of lateral cell wall structure. Scale bar represents 1 µm.

**Figure 5 plants-10-00734-f005:**
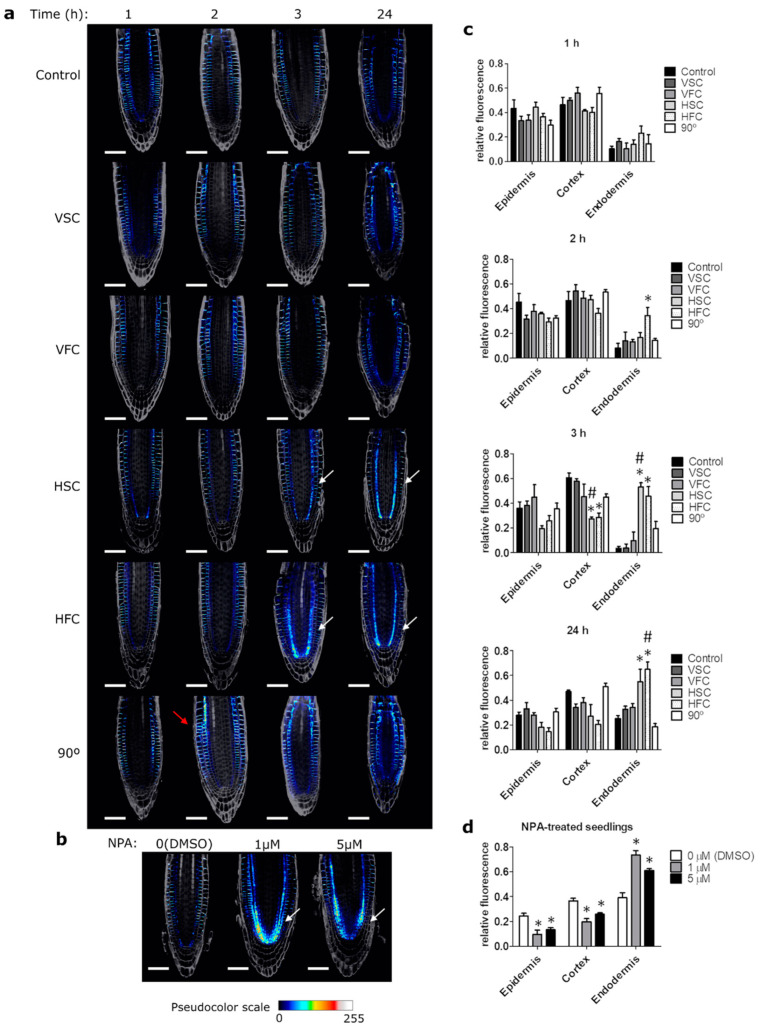
PIN-FORMED2 (PIN2) distribution and levels in the root meristem. (**a**) Confocal microscope images of PIN2-Green Fluorescence Protein (GFP) reporter line. Pseudocolor reflecting the intensity of the GFP signal was applied with the Lookup Table Royal Tool in the ImageJ software; Grey: cell wall staining with Renaissance SR2200. Experimental conditions: Control; VSC, Vertical Slow Clinorotation; VFC, Vertical Fast Clinorotation; HSC, Horizontal Slow Clinorotation; HFC, Horizontal Fast Clinorotation; 90°, directional growth control. At times 1, 2, 3, and 24 h of exposure to these conditions. White arrows highlight endodermis accumulation. Red arrow: PIN2 redistribution to new gravity vector. Scale bar represents 50 µm (**b**) PIN2-GFP seedlings grew for 5 days with 0 (DMSO), 1 or 5 µM naphthylphthalamic acid (NPA). Scale bar represents 50 µm. (**c**) Relative fluorescence GFP intensity in the different meristem cell layers at the different times and experimental conditions. (**d**) GFP relative intensity at each meristem layer in seedlings grown for 5 days with 0 (Dimethyl Sulfoxide, DMSO), 1 or 5 µM NPA. * *p*-value < 0.05 compared to control at the same time and layer (or 0 µm DMSO in the NPA-treated seedlings), # *p*-value < 0.05 compared to the same time and clinorotation speed but different orientation.

**Figure 6 plants-10-00734-f006:**
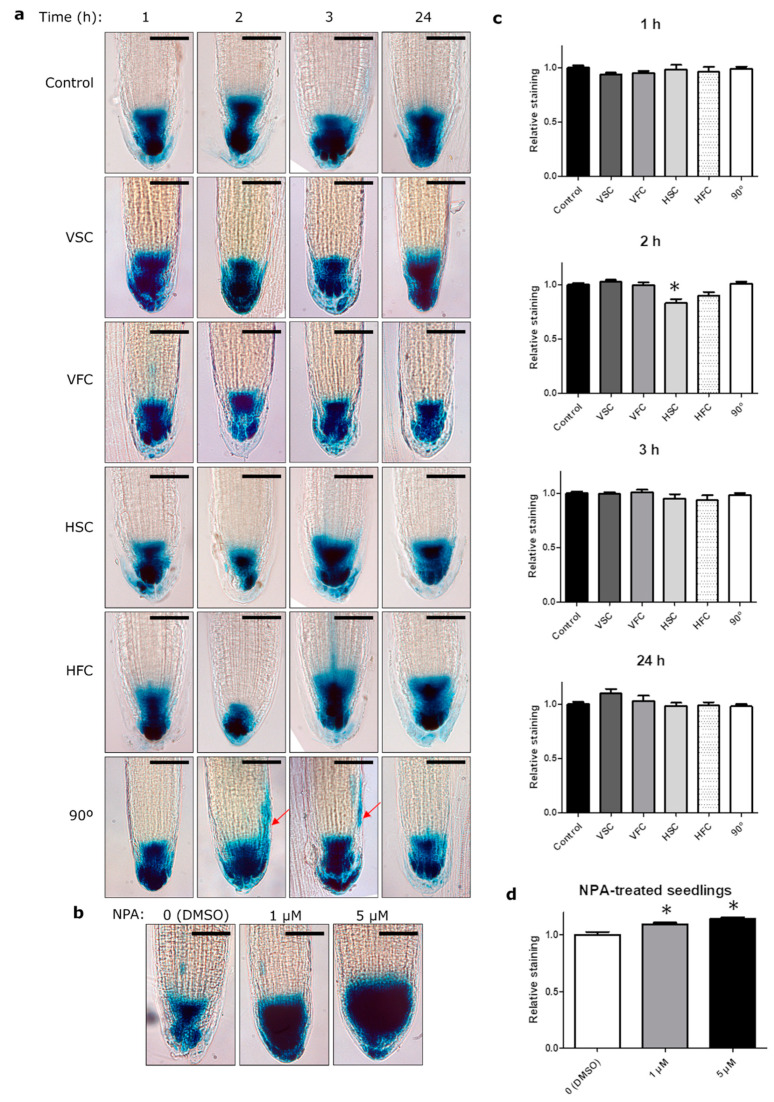
Auxin distribution among clinorotation conditions. (**a**) Optical microscope images of DR5-β-glucuronidase (DR5-GUS) reporter line seedlings exposed to the different experimental conditions: Control; VSC, Vertical Slow Clinorotation; VFC, Vertical Fast Clinorotation; HSC, Horizontal Slow Clinorotation; HFC, Horizontal Fast Clinorotation; 90°, directional growth control; after 1, 2, 3, or 24 h of exposure. Red arrows indicate auxin redistribution. Scale bar represents 50 µm (**b**) DR5-GUS seedlings grew for 5 days with 0 (DMSO), 1 or 5 µM NPA. Scale bar represents 50 µm. (**c**) DR5 relative staining quantification at the different times (1, 2, 3, and 24 h) and conditions. (**d**) NPA-treated seedlings DR5-GUS relative staining quantification. Bars represent mean + SEM. * *p*-value < 0.05.

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
