# Peer review of "Analysis of Graviresponse and Biological Effects of Vertical and Horizontal Clinorotation in Arabidopsis thaliana Root Tip"

_plants, 2021, doi:10.3390/plants10040734_

Round 1

Reviewer 1 Report

In this manuscript, Villacampa et al. report their comparative analyses of root gravitropic response of Arabidopsis among different 2D-clinorotating conditions.  Their results showed that root growth behaviours under clinorotating conditions differed from either control or gravistimulated ones.  However, root growth tended to be affected by centrifugal force, which was particularly obvious in plants under fast rotation.  Their microscopic analyses showed that amyloplasts that reside in root columella cells showed dispersed distribution under slow rotating condition.  In addition, their ultrastructural observation revealed that columella cell walls located at distal position were deformed when plants were rotated horizontally.  They also observed localization of PIN2 protein, an indispensable auxin efflux carrier for gravitropism, as well as auxin distribution.  They found that in horizontally rotated roots, PIN2 accumulated much higher in endodermis that any other samples.  From these results, they concluded that vertical, slow, vertical rotating condition seemed to be the best condition to lessen the artifact.  As the clinorotation is the most feasible way to mimic microgravity on Earth, this report has a certain importance.  The results are quite clear and their logic is easy to follow.  However, there remains some concerns that the authors should be addressed.

My major concern is as follows;

  1. Their main conclusion is mainly based on microscopic observation. In this context, I wonder how they fixed the clinorotating samples. They should be fixed under clinorotating conditions, however there are no descriptions in the manuscript.  Otherwise, the fixation conditions should be taken account into their discussion, if they are fixed under stationary condition. 
  2. From their electron micrographs, they mentioned that some organelles, namely mitochondria, amyloplasts and endosomes are deformed under horizontally rotated samples. However, an electron micrograph shows just a section of a certain organelle and it does not visualize the entire shape of it. If the authors wish to emphasize the deformation of organelles, they should analyse a series of sections by 3D-reconstruction.
  3. In their confocal microscopic observation, they found the alteration of PIN2 expression in endodermis of horizontally rotated roots. As far as I aware, PIN2 is expressed mainly in epidermis and cortex. In addition, analyses of endodermis deficient mutants showed that it is nor required for gravitropism in roots.  The authors should discuss the biological meaning of enhanced expression of PIN2 under such condition.

Other minor comments;

  1. Some citations are mentioned as “xxx et al., yyyy”, e.g. lines 32-33, 162 and 285. And they are all missing in the reference list. I am not sure whether such type of description is in agreement with the instruction, nevertheless I strongly recommend the authors to include them to reference list. 
  2. In each statistical analysis, authors should describe the methods and the number of the samples. I do not think that t-test is appropriate, rather I recommend to use multiple comparison procedure.

3. Line 215: “HSC” should be “HFC”.

Author Response

In the reviewer´s report concerning the proposition for publication of the manuscript entitled “Analysis of graviresponse and biological effects of vertical and horizontal clinorotation in Arabidopsis thaliana root tip” in a special issue of “Plants”: “Root Tropisms: New Insights Leading the Growth Direction of the Hidden Half” the reviewer expresses few major and minor concerns that we would like to address below. Line numbers correspond to the corrected document when all of the introduced changes are visible (“All Markup” option in Word).

Major concerns:

  1. Their main conclusion is mainly based on microscopic observation. In this context, I wonder how they fixed the clinorotating samples. They should be fixed under clinorotating conditions, however there are no descriptions in the manuscript. Otherwise, the fixation conditions should be taken account into their discussion, if they are fixed under stationary condition.

The description of the fixation procedure was added (L557-561; L577-578). Samples were fixed immediately after the clinorotation, taking care of minimizing the time elapsed between the arrest of the clinostat and the interaction of the fixative with samples. First, the fixative was directly added to the Petri dish just after the release from the clinostat, for an immediate arrest of the vital activity. Then, seedlings were transferred to centrifuge tubes filled with fixative. Since the procedure was always performed in the same manner and we have observed clear differences between the conditions, we concluded that the fixation procedure did not significantly influence the results. We have added a short paragraph discussing the impact of the fixative procedure on the results (L239-242).

  1. From their electron micrographs, they mentioned that some organelles, namely mitochondria, amyloplasts and endosomes are deformed under horizontally rotated samples. However, an electron micrograph shows just a section of a certain organelle and it does not visualize the entire shape of it. If the authors wish to emphasize the deformation of organelles, they should analyse a series of sections by 3D-reconstruction.

We have considered the reviewer´s concern and we agree that a doubtless conclusion on the organelle deformation can only be reached after 3D ultrastructural reconstruction. However, performing a 3D-reconstruction from micrographs obtained from the Transmission Electron Microscope is a very complex and laborious process. As surely known by the reviewer, the procedure requires the attainment of a perfect ribbon formed by a significant number of consecutive ultrathin sections, which should be mounted in a special (highly delicate) support of transparent film, devoid of any gridded structure that could interfere with the observation in each one of the sections of the ribbon. Each ultrathin section is 80–100 nm thick. To perform the reconstruction, we would have to gather consecutive micrographs of each organelle from each section, which should be processed with the appropriate software in order to get the 3D structure. In our opinion, that we consider realistic, this procedure is only worth to be carried out if the result to obtain is central and essential for the objectives of the research work and for the demonstration of the main work hypothesis. In our case, we think that this result is certainly important and worth to be mentioned in the paper, but it does not certainly constitute the major conclusion to fulfill our work hypothesis, as the most important indication of the mechanical stress is the cell wall deformation in horizontally clinorotated samples. For this reason, we keep the result in the paper, but reduce the strength of the statement, leaving it as a strong suggestion, and not a doubtless conclusion (L243, L251, L253-254, L440). We have edited the section 2.3. to be more precise in describing the differences we have observed between the conditions. To determine the described changes, we have compared a set of micrographs for each condition and have observed a repetitive pattern. The micrographs that are included in the figures are representative images for each condition. 

  1. In their confocal microscopic observation, they found the alteration of PIN2 expression in endodermis of horizontally rotated roots. As far as I aware, PIN2 is expressed mainly in epidermis and cortex. In addition, analyses of endodermis deficient mutants showed that it is not required for gravitropism in roots. The authors should discuss the biological meaning of enhanced expression of PIN2 under such condition.

In samples exposed to vertical clinorotation, as well as in control samples, we have detected the typical localization of PIN2 protein in the epidermis and cortex. This localization was also observed in the horizontally clinorotated samples, but, additionally, an accumulation of this protein in the endodermis was detected in the samples clinorotated for 3 hours or more. We consider that this accumulation is more likely to be related to the mechanical stress that the seedlings are experiencing during prolonged horizontal clinorotation, rather than to the gravitropic stimulation. A similar PIN2 localization in the endodermis was observed in the crk5-1 mutant that is deficient in CRK5. This protein is a plasma membrane–associated kinase, which phosphorylates the hydrophilic loop of PIN2. We have added a paragraph discussing in more detail this PIN2 accumulation in the endodermis (L454-472).

Minor concerns:

  1. Some citations are mentioned as “xxx et al., yyyy”, e.g. lines 32-33, 162 and 285. And they are all missing in the reference list. I am not sure whether such type of description is in agreement with the instruction, nevertheless I strongly recommend the authors to include them to reference list.

We have corrected the format and included the references in the bibliography list (L32-33, L182, L311).

  1. In each statistical analysis, authors should describe the methods and the number of the samples. I do not think that t-test is appropriate, rather I recommend to use multiple comparison procedure.

The number of samples and methods of statistical analysis for each quantification were given in the section 4.4 (L601-603, L605-607).

  1. Line 215: “HSC” should be “HFC”.

 We have corrected the mistake (L237).

Reviewer 2 Report

In this manuscript, Villacampa et al. compared the effects of slow vs. fast clinostat and vertical vs. horizontal clinostats on root gravitropism and potential stress caused by this. Based on the results, they proposed that the slow clinostat with around 1 rpm is enough to mimic the microgravity environment, which is indicated by that statoliths are dispersed throughout the columella cells rather than located at the bottom of the cells. However, the fast clinostat with 60 rpm results in the directional growth of roots due to the centrifugal force. These results suggest that slow clinorotation is better to mimic microgravity as compared to fast clinorotation. 

Additionally, both the vertical and horizontal clinostat can be used to mimic the microgravity field. However, in comparison to the vertical clinostat, the horizontal clinostat generates the stress response and mechanostimulation, as indicated by the changes in statocyte ultrastructure and enhanced internalization of the PIN2 auxin transporter under horizontal clinorotation. These results suggest that the vertical clinorotation is superior to the horizontal clinorotation in simulating the field of microgravity.

In summary, by comparing different settings of clinostat systematically, for the first time, they showed that the vertical slow clinostat (VSC) is the optimal choice for simulating the microgravity for Arabidopsis growth. Generally, these results are interesting, the data support the conclusion, the paper is well written. I have very few comments for authors to address and improve the draft.

  1. I strongly suggest the author put Fig. 6 in Fig. 1. For Fig. 6, this schematic diagram perfectly shows the audience what are the vertical and horizontal clinostats. At the beginning of this draft, the author used some paragraphs to introduce the setting of vertical and horizontal clinostats without any figures, it is very hard to be understood. Meanwhile, in Line 149, the author cites Figure 6 after Fig.1 by skipping other figures, it is not so rational.
  2. Line 53, the word ‘phytohormone’ is ambiguous here, replaced it with ‘auxin’.
  3. Line 113, change “the root growth according to the gravity vector” to “the root growth toward gravity vector”.
  4. Line 191-201, for this paragraph, there are no data at all, the author can consider integrating this paragraph into the Introduction section.
  5. Line 212, please define the “FC”.
  6. Line192, actually the layer of the cells closest to the quiescent center is the CSC, it is not the S1, which only denotes the first layer of columella cell below the CSC.

Author Response

In the reviewer´s report concerning the proposition for publication of the manuscript entitled “Analysis of graviresponse and biological effects of vertical and horizontal clinorotation in Arabidopsis thaliana root tip” in a special issue of “Plants”: “Root Tropisms: New Insights Leading the Growth Direction of the Hidden Half” the reviewer expresses few minor concerns that we would like to address below. Line numbers correspond to the corrected document when all of the introduced changes are visible (“All Markup” option in Word).

  1. I strongly suggest the author put Fig. 6 in Fig. 1. For Fig. 6, this schematic diagram perfectly shows the audience what are the vertical and horizontal clinostats. At the beginning of this draft, the author used some paragraphs to introduce the setting of vertical and horizontal clinostats without any figures, it is very hard to be understood. Meanwhile, in Line 149, the author cites Figure 6 after Fig.1 by skipping other figures, it is not so rational.

The figure presenting the modes of clinorotation was moved to the Results section, as advised (L113-L123).

  1. Line 53, the word ‘phytohormone’ is ambiguous here, replaced it with ‘auxin’.

“Phytohormone” was replaced with “auxin” (L63)

  1. Line 113, change “the root growth according to the gravity vector” to “the root growth toward gravity vector”.

The phrase was replaced as indicated (L139).

  1. Line 191-201, for this paragraph, there are no data at all, the author can consider integrating this paragraph into the Introduction section.

The paragraph describing columella structure and function (in new document L212-220) was moved to the introduction (L38-L49).

  1. Line 212, please define the “FC”.

FC has been defined as indicated (L234).

  1. Line192, actually the layer of the cells closest to the quiescent center is the CSC, it is not the S1, which only denotes the first layer of columella cell below the CSC.

The mistake was corrected and the “Columella Stem Cells” term (CSC) was introduced (L41-43).